# In Vivo Efficacy and Toxicity of an Antimicrobial Peptide in a Model of Endotoxin-Induced Pulmonary Inflammation

**DOI:** 10.3390/ijms24097967

**Published:** 2023-04-27

**Authors:** Laura Cresti, Giovanni Cappello, Silvia Vailati, Elsa Melloni, Jlenia Brunetti, Chiara Falciani, Luisa Bracci, Alessandro Pini

**Affiliations:** 1U.O.C. Clinical Pathology, Azienda Ospedaliera Universitaria Senese, Via M. Bracci, 53100 Siena, Italy; 2Medical Biotechnology Department, University of Siena, Via A Moro 2, 53100 Siena, Italy; 3SetLance srl, Via Fiorentina 1, 53100 Siena, Italy; 4Zambon spa, Via A. Meucci 3, 20091 Bresso, Italy

**Keywords:** antimicrobial peptide, anti-inflammatory efficacy, in vivo toxicity, NOAEL, pulmonary inflammation

## Abstract

SET-M33 is a synthetic peptide that is being developed as a new antibiotic against major Gram-negative bacteria. Here we report two in vivo studies to assess the toxicity and efficacy of the peptide in a murine model of pulmonary inflammation. First, we present the toxicity study in which SET-M33 was administered to CD-1 mice by snout inhalation exposure for 1 h/day for 7 days at doses of 5 and 20 mg/kg/day. The results showed adverse clinical signs and effects on body weight at the higher dose, as well as some treatment-related histopathology findings (lungs and bronchi, nose/turbinates, larynx and tracheal bifurcation). On this basis, the no observable adverse effect level (NOAEL) was considered to be 5 mg/kg/day. We then report an efficacy study of the peptide in an endotoxin (LPS)-induced pulmonary inflammation model. Intratracheal administration of SET-M33 at 0.5, 2 and 5 mg/kg significantly inhibited BAL neutrophil cell counts after an LPS challenge. A significant reduction in pro-inflammatory cytokines, KC, MIP-1α, IP-10, MCP-1 and TNF-α was also recorded after SET-M33 administration.

## 1. Introduction

The overuse and misuse of antibiotics in healthcare, agriculture and veterinary practice have led to noticeably increased antimicrobial resistance [1,2]. This is a huge concern, since most areas of modern medicine are unthinkable without effective antimicrobial treatment [3].

Unfortunately, the increase in antibiotic-resistant bacteria cannot be countered by the development of drugs with a new mode of action, which is currently poor and ineffective [4]. Only about 30–40 new antibacterial compounds are currently in the clinical trial phases [5], and those targeting World Health Organization (WHO) priority pathogens are derivatives of existing classes [6]. Only a minority of the antibiotics approved over the past 40 years belong to new drug classes, while the majority are derived from known chemical structures, and the most recent new class of antibiotics was discovered as long ago as the 1980s [7]. Strategic global investment in new therapeutic options to fight antimicrobial resistance is therefore urgently needed [8]. Antimicrobial peptides are considered an interesting class of antibacterial molecule [9,10]. Although they are not a complete alternative to traditional antibiotics due to certain issues, they could be a valid support to antibiotic therapy [11,12,13,14,15,16,17].

Peptides in general have become an increasingly important therapeutic class. Although the field of therapeutic peptides started with natural hormones, the discovery and development trends have shifted from simply mimicking natural hormones or peptides derived from nature to the rational design of peptides with desirable biochemical and physiological properties [18]. Major breakthroughs in molecular biology, peptide chemistry and peptide delivery technologies have allowed significant progress in the field of peptide drug discovery, peptide production and their therapeutic applications [19,20,21]. Indeed, over the years, the demand for peptide therapeutics has increased substantially, leading to more than 100 peptide drug approvals [22]. However, the prime focus has remained on anticancer and antimicrobial therapeutics [23]. 

SET-M33 is a non-natural antimicrobial peptide with a promising efficacy profile for various clinical applications. Its tetra-branched form confers high resistance to protease and peptidase activities, making it a good candidate for in vivo use [24,25,26]. The peptide is active against a panel of clinically important Gram-negative bacteria [27], and it has been characterized in preclinical stages for efficacy (sepsis, lung infections, skin infections), full toxicity, bio-distribution, excretion, selection of resistances, gene toxicity, mechanism of action, immunomodulatory activity and time-kill concentrations [28,29,30,31]. Data on analogous forms of the peptide produced with D-amino-acids [32], in dimeric form [33], in a pegylated version [34], encapsulated in dextran nanoparticles [35] and in poly(lactide-co-glycolide) nanoparticles [36] are already reported in the literature. The peptide SET-M33 has completed preclinical development with toxicity results for intravenous administration in rats and dogs, two animal species recommended as rodent and non-rodent test systems, respectively, by international guidelines [37]. 

In view of its possible use as a new antimicrobial drug administered by aerosol for lung infections, we report: (1) the local and systemic toxicity of SET-M33 in a 1-week inhalation study in CD-1 mice; (2) the efficacy of SET-M33 aerosolized in a murine model of endotoxin (LPS)-induced pulmonary inflammation.

## 2. Results and Discussion

### 2.1. Toxicity Study by Inhalation Administration to CD-1 Mice for a Week

In this study we evaluated the toxicity of SET-M33 during a one-week-inhalation study in CD-1 mice. The mice (six males and six females/group) were treated with the peptide at 5 mg/kg/day, 20 mg/kg/day or vehicle only (control) by inhalation every day for 1 week. These doses were selected by preliminary studies on SET-M33 efficacy and toxicity [33,36].

#### 2.1.1. Atmosphere Analysis and Estimation of Achieved Dose

An aerosol administration study was performed. Atmosphere analysis, including data on the concentration of peptide achieved in the aerosol, the particle size distribution in the aerosol and the estimated dose of peptide inhaled by the mice is reported in Table 1. The aerosol concentrations achieved were 87% and 117% of the target concentrations (Table 1), and the estimated inhaled doses achieved were 97% and 116% of the target doses for animals treated at 5 and 20 mg/kg/day, respectively. The mass median aerodynamic diameter (MMAD) of the droplets was slightly below for the lower dose and within the ideal range (1–3 µm) of a repeated dose inhalation study for the higher dose. For technical details and the formula used to calculate the estimated inhaled dose, see Material and Methods.

From now on throughout the article, the doses administered will be indicated as 4.34 mg/kg/day and 23.1 mg/kg/day.

#### 2.1.2. Clinical Observations

During the administration period, the animals were inspected visually at least twice daily for evidence of ill health or reaction to treatment. Clinical condition, body weight, food consumption, organ weight, macropathology, histopathology and toxicokinetic were observed. The weight of each animal was recorded twice during the study and before necropsy.

There were no mortalities. Clinical signs in animals given 23.1 mg/kg/day were noted after the first dose. These included decreased activity, unsteadiness, cold to touch, partially closed eyelids, tremor, piloerection, hunched posture, splayed limbs and irregular breathing. The incidence and duration of the signs lessened as the study progressed. Regarding body weight, in males dosed at 23.1 mg/kg/day, group mean body weight loss (−2.2%) exceeded the loss seen in the control group (−0.32%) over days 1–8. In females, mean body weight loss was slightly less (−2.65%) than that of the control group (−4.8%). Group mean food consumption in males at 23.1 mg/kg/day was slightly less than controls (−16%). A similar effect was not seen in females. There were no effects on body weight or food consumption in animals dosed with 4.34 mg/kg/day. There was no restriction on diet supply. The weight of food supplied to each cage was recorded for the week before the treatment started and for each week throughout the study. 

#### 2.1.3. Necroscopy

In animals dosed with 23.1 mg/kg/day, group mean body weight adjusted lung and bronchi weight was higher than the controls in males and females (+5.6% and +25.61%, respectively), group mean body weight adjusted liver weight was lower than the concurrent controls in males (* *p* < 0.05) and females (−11.6% and −13.37%, respectively) and group mean body weight adjusted kidney weight was statistically significantly (* *p* < 0.05) lower than the control for males (−12.94%). In animals dosed with 4.34 mg/kg/day, group mean body weight adjusted lung and bronchi weight was higher than the control (+5.9% and +12% for male and females, respectively).

Interstitial and granulomatous inflammation associated with perivascular infiltrate of inflammatory cells and/or alveolar inflammation was seen in the lungs of animals given 23.1 mg/kg/day, with fibrosis of the alveolar ducts in one female. Minimal interstitial inflammation was seen in two females on 4.34 mg/kg/day. Atrophy/degeneration of the olfactory epithelium and minimal-to-marked inflammatory exudate was seen in the nose/turbinates of all treated animals. In one male on 23.1 mg/kg/day, the exudate was haemorrhagic. These changes were generally associated with an increase in eosinophilic globules in the olfactory and respiratory epithelium and inflammatory cell infiltrate in the lamina propria. Minimal squamous metaplasia of the respiratory epithelium was seen in the larynx of animals on 4.34 or 23.1 mg/kg/day and was associated with minimal inflammatory cell infiltrate in most animals on 23.1 mg/kg/day and three males on 4.34 mg/kg/day. Loss of respiratory epithelial cilia at the tracheal bifurcation was observed in animals on 23.1 mg/kg/day. Macroscopically enlarged tracheobronchial lymph nodes were seen in two females on 23.1 mg/kg/day. The above changes were considered adverse only at 23.1 mg/kg/day. A summary of all these findings is reported in Table 2.

#### 2.1.4. Bioanalysis

Blood samples were obtained from 2 animals/group/sex at day 7, immediately and 23 h after dosing. Plasma concentrations of SET-M33 were only quantifiable in 1 out of 2 males (20.3 ng/mL) and both females (17.7 and 23.1 ng/mL) immediately after dosing at an achieved inhaled dose of 23.1 mg/kg. No quantifiable levels of the peptide were detected 23 h after dosing. 

### 2.2. Efficacy of SET-M33 on Endotoxin (LPS)-Induced Lung Inflammation

Vehicle (0.9% saline solution) or 0.5, 2, 5 mg/kg SET-M33 or 1 mg/kg budesonide (positive control) was administered intratracheally to male BALB/c mice (10 animals/group) approximately 30 min prior to saline or the LPS challenge (0.5 mg/mL). Budesonide is a corticosteroid drug administered clinically by inhalation to reduce inflammation of the lungs in cases of acute asthma [38]. The animals were sacrificed 4 h after the LPS challenge, and bronchoalveolar lavage (BAL) was performed. Total and differential cell counts were carried out in the BAL fluid recovered from the mice, and levels of cytokines IL-6, IP-10, KC, MCP-1, MIP-1α and TNF-α were quantified in BAL supernatant. There were no mortalities. Clinical signs showed dose dependency in severity and onset; however, all animals in groups dosed at 2 and 5 mg/kg showed clinical signs 1 h post-dose (immediately after the challenge). 

#### 2.2.1. Total and Differential Cell Counts in BAL 

The LPS challenge resulted in a statistically significant increase in the BAL total cell count, neutrophils, eosinophils and lymphocytes and a significant decrease in mononuclear cells compared with vehicle-treated saline-challenged mice (Figure 1). Intratracheal administration of SET-M33 at 0.5 and 2 mg/kg produced a statistically significant decrease in the BAL total cell count (Figure 1A), neutrophils (Figure 1B) and lymphocytes (Figure 1E), a significant increase in mononuclear cells (Figure 1D) and no significant effect on eosinophils (Figure 1C) compared with the vehicle-treated LPS-challenged mice. SET-M33 at 5 mg/kg produced a statistically significant decrease in neutrophils (Figure 1B), a significant increase in mononuclear cells (Figure 1D) and no significant effect on the BAL total cell count (Figure 1A), lymphocytes (Figure 1E) and eosinophils (Figure 1C) compared with vehicle-treated LPS-challenged mice. The maximum inhibition of neutrophil cell infiltrate was recorded at all three doses of SET-M33 (Figure 1B), which resulted in an equivalent to steroid efficacy, but a dose-dependent increase in BAL mononuclear cells (macrophages) was recorded after the administration of SET-M33 (0.5, 2 and 5 mg/kg) (Figure 1D). Budesonide at 1 mg/kg produced a significant decrease in neutrophils (Figure 1B) and a marked decrease in the BAL total cell count, which did not quite reach statistical significance with respect to vehicle-treated LPS-challenged mice (Figure 1A).

#### 2.2.2. BAL Cytokine Levels

The LPS challenge resulted in a statistically significant increase in the BAL concentrations of IL-6, IP-10, KC, MCP-1, MIP-1α and TNF-α compared with vehicle-treated saline-challenged mice (Figure 2). SET-M33 at 0.5 mg/kg produced significant inhibition of IP-10 (Figure 2B), MCP-1 (Figure 2D) and MIP-1α (Figure 2E) and no significant effect on IL-6 (Figure 2A), KC (Figure 2C) or TNF-α (Figure 2F). The same was found for SET-M33 at 2 mg/kg (Figure 2A–E) except that TNF-α was significantly inhibited (Figure 2F). SET-M33 at 5 mg/kg produced significant inhibition of five cytokines (IP-10, KC, MCP-1, MIP-1α and TNF-α; Figure 2B–F), and a significant increase in IL-6 (Figure 2A). The inhibitory effect of SET-M33 on pro-inflammatory cytokines was clearly dose-dependent for KC, TNF-α and MIP-1α (Figure 2C,E,F). Inhibition of IP-10 and MCP-1 after administration of SET-M33 at all doses was similar to that recorded after budesonide. SET-M33 administration at 5 mg/kg proved more effective than steroid (budesonide 1 mg/kg) in inhibiting MIP-1α and TNF-α. Budesonide 1 mg/kg significantly inhibited all the cytokines tested. 

## 3. Materials and Methods

### 3.1. Peptide SET-M33

Peptide with a purity of 96% as declared by the producer (Polypeptide, Strasbourg, France) was used for all tests. The formulations were prepared under sterile conditions by dissolving the powder in water at appropriate concentrations and then sterile filtering with a 0.22 micron PVDF filtration unit. Dose formulations of SET-M33 and water were analyzed to confirm that the dose formulations prepared were homogeneous and that the SET-M33 concentrations administered were appropriate under the study conditions. The analytical method validated at the Testing Facility involved dilution of SET-M33 dose formulation samples in 100% water followed by quantification using high performance liquid chromatography with ultraviolet detection.

### 3.2. Animals and Experimental Procedures

Crl:CD-1(ICR) mice and male BALB/c mice were used for the toxicity and efficacy studies, respectively. All animals were supplied by Charles River Ltd. (Margate, UK). After 10 days of acclimation prior to the first day of dosing, the animals were randomized into study groups, identified and allocated into monitored cages.

For the toxicity study, administration was by inhalation—snout-only exposure using a specific inhalation exposure system for mice (further details about the inhalation exposure system are reported in Appendix A). Animals were dosed once daily for 7 days. The duration of the exposure was 60 min each day.

For the efficacy study, animals were dosed once with vehicle, SET-M33 or budesonide intratracheally approximately 30 min prior to the beginning of the LPS challenge. Animals were weighed on the day of dosing and a fixed volume of 50 µL was administered (SET-M33 at 0.2, 0.8, 2 and mg/mL). They were anaesthetized with a mix of isoflurane/oxygen (administered by inhalation at the following concentrations: 3–4% for induction and 1–3% for maintenance) and secured to an intubation device with a cord around the upper incisors. The tongue was pulled forward and any excess food and/or mucus was removed. A veterinary operating otoscope was inserted into the animal’s mouth to illuminate the posterior pharynx and epiglottis. The tip of the dosing device was guided through the vocal cords into the lumen of the trachea. The pipette tip was guided down the trachea and the pipette plunger depressed with firm pressure. After dosing, the animals were removed from the secured position and carefully monitored until full recovery.

For the inflammatory challenge, lipopolysaccharide (LPS) from *Escherichia coli* serotype O26:B6 was used (#L3755, Sigma-Aldrich^®^, St. Louis, MO, USA). Animals were placed in groups in an acrylic box and 8 mL LPS in 0.9% *w*/*v* saline was placed in each of two jet nebulizers (Sidestream^®^, Philips N.V. Amsterdam, The Netherlands). Compressed air at approximately 6 L/min was passed through each nebulizer and the output of the nebulizers directed into the box containing the animals for 30 min. On completion of the inflammatory challenge, the mice were returned to their home cage. Budesonide (#B7777, Sigma-Aldrich^®^) was used as a positive control. It was prepared in 0.1% Tween 80/0.6% NaCl in 0.05 M pH 6.0 phosphate buffered saline. 

All animal experiments were approved by the local ethics committee of CRO, where the experiments were performed: Covance CRS LLC (now Labcorp Drug Development), Huntingdon, Cambridgeshire, UK. All experimental procedures during the studies were subject to the provisions of the United Kingdom Animals (Scientific Procedures) Act 1986 Amendment Regulations 2012 (the Act). All methods are reported according to ARRIVE guidelines [39]. The number of animals used was the minimum consistent with scientific integrity and regulatory acceptability, considering the welfare of individual animals in relation to the number and extent of procedures to be carried out on each animal.

### 3.3. Toxicity Study by Inhalation Administration to CD-1 Mice for 1 Week

#### 3.3.1. Atmosphere Analysis and Estimation of Achieved Dose

The concentration in the aerosol and the particle size distribution were analyzed by ultra-performance liquid chromatography. Briefly, the sample was collected from the filtration units of the aerosol machine, extracted with extraction solvent (0.1% trifluoroacetic acid and 0.05% Tween 80 in water, #302031 and #8.22187, respectively, supplied by Sigma-Aldrich^®^) and injected into the chromatograph. The chromatograph system was calibrated using an external standard. Peak area data acquired by the data capture software underwent least squares regression analysis and the concentration was calculated. 

Mass Median Aerosol Diameter (MMAD) and Geometric Standard Deviation (GSD) were calculated by linear regression of the probit of the cumulative percentage, by mass, of particles smaller than the effective cut-off diameter of each stage versus the logarithm of the cut-off diameter of each stage. Particle size was determined by cascade impaction using a Marple 290 series (296 configuration) cascade impactor at a flow rate of 2.0 L/min.

The estimations of inhaled dose (mg/kg/day) were calculated using the formula [40]: $$\mathrm{Dose}\;(\mathrm{mg}/\mathrm{kg}/\mathrm{day})=\frac{C\;\left(\mu g/L\right)\times \;\mathrm{RMV}\;\left(L/\min\right)\times \;D\;\left(\min\right)}{\mathrm{BW}\;\left(\mathrm{Kg}\right)\times 1000\;}$$

where C = aerosol concentration (μg/L)RMV = respiratory minute volume = 0.608 × BW (Kg)^0.852^D = duration of exposure (60 min)BW = body weight (kg).

#### 3.3.2. Necroscopy

Overdose of intraperitoneal pentobarbital sodium (at 50 mg/kg) followed by exsanguination was used for termination. All animals underwent detailed necropsy. A full macroscopic examination of the tissues was performed. Any abnormality in the appearance or size of any organ or tissue was recorded and the required tissue samples preserved in appropriate fixative. The organs were weighed. Tissue samples were routinely preserved in 10% Neutral Buffered Formalin (except testes and eyes, which were preserved in modified Davidson’s fluid Eyes and Davidson’s fluid, respectively). Tissue samples were dehydrated, embedded in paraffin wax and sectioned at a nominal four- to five-micron thickness. For bilateral organs, sections of both organs were prepared. A single section was prepared from each of the remaining tissues required. Sections were stained with haematoxylin and eosin. Each section was examined microscopically. Findings were either reported as “present” or assigned a five-point severity grade: minimal, slight, moderate, marked or severe. 

#### 3.3.3. Bioanalysis

Blood samples were collected from the jugular vein under isoflurane anaesthesia. Each sample was collected into a polypropylene test tube containing lithium heparin as anticoagulant and centrifuged at 2000× *g* for 10 min at +4 °C. Total concentrations of peptide were determined by a qualified LC-MS/MS method. No specific testing regulations or guidelines were applicable for the plasma sample analysis; however, the analytical procedures followed were based on [41]. All data collection and processing were performed by the data acquisition system associated with the mass spectrometer (PC Analyst V1.6.1 or later). Data processing and quantification were performed using a Watson LIMS V7.2. After analysis, samples were immediately stored at −80 °C. 

### 3.4. Efficacy of SET-M33 in a Murine Model of Endotoxin (LPS)-Induced Pulmonary Inflammation

Following confirmation of death, an incision was made in the neck and the muscle layers separated by blunt dissection to isolate the trachea. A small incision was made in the trachea and a tracheal cannula inserted. The cannula was secured in place and the airway was washed with 0.3 mL phosphate buffered saline (PBS). The PBS was left in the airway for approximately 10 s while the chest was gently massaged and then removed. This was repeated twice more. In total, three lots of 0.3 mL PBS were used for lung lavage. The BAL fluid from the three lavages was pooled and placed on wet ice until centrifuged. Each BAL sample was centrifuged at 1000× *g* for 10 min at approximately 4 °C. The cell pellet was resuspended in 0.5 mL PBS and stored on wet ice prior to analysis. A total and differential cell count of the re-suspended BAL cells was performed using the XT-2000iV (Sysmex UK Ltd., Milton Keynes, UK.). The sample was vortexed for approximately 5 s and analyzed. Total and differential cell counts (including neutrophils, lymphocytes, eosinophils and mononuclear cells (includes monocytes and macrophages) are reported as number of cells per animal. BAL cytokines TNF α, IL-6, MIP-1α, KC, MCP-1 and IP-10 were analyzed using a multiplex system following the manufacturer’s protocol (#LXSAMSM, Biotechne^®^ Mouse Magnetic Luminex Screening Assay, R&D System, Minneapolis, MN, USA).

### 3.5. Statistical Analysis

For organ weight data in toxicity study, analysis of covariance with Student’s *t* test was performed using terminal body weight as covariate [42], unless non-parametric methods were applied. The treatment comparisons were made on adjusted group means in order to allow for differences in body weight that might influence the organ weights. Significant differences between the groups compared were calculated using GraphPad Prism for Windows version 5.03, GraphPad Software, San Diego, CA, USA, and expressed as * *p* < 0.05. 

Regarding efficacy study, data were expressed as means ± standard errors of the means. Statistical analysis was used to determine whether there were significant differences in BAL total and differential cell counts and BAL cytokine levels between SET-M33 (0.5, 2 and 5 mg/kg) or budesonide (1 mg/kg)-treated LPS-challenged animals and vehicle-treated LPS-challenged animals. Statistical analysis was also performed to compare vehicle-treated LPS-challenged animals with vehicle-treated saline-challenged animals. Statistical analysis was performed using the Student’s *t*-test with GraphPad Prism for Windows version 5.03, GraphPad Software, San Diego, CA, USA. The levels of statistical significance are indicated in the legends of the figures.

## 4. Conclusions

Pneumonia is a common acute respiratory infection that affects the alveoli and distal airways [43]. It is a major health problem and is associated with high morbidity and short- and long-term mortality in all age groups worldwide. A large variety of microorganisms can cause pneumonia, including multi-drug resistant pathogens such as *S. aureus*, *P. aeruginosa*, *K. pneumoniae* and other bacteria [44,45]. Research focused on new antibacterial drugs to be administered locally into the lungs is of crucial importance for the future. Pneumonia is triggered by bacterial infections that evolve into an inflammatory state, in some cases very severe, in which bacterial toxins such as LPS, LTA, peptidoglycan and others play an important role. However, LPS is the major toxin involved in lung inflammation in pneumonia [46]. The neutralization of LPS and the killing of bacteria are therefore important therapeutic issues for healthcare professionals.

The antimicrobial peptide SET-M33 has been extensively described in terms of antibacterial activity against pathogenic species of high clinical interest. Here, we described the activity of SET-M33 in inhibiting the pro-inflammatory action of LPS in the lung through local administration by nebulization in mice.

The therapeutic effect of the peptide was evaluated in terms of inhibition of the cells and cytokines involved in inflammation. The strongest activity of SET-M33 was recorded for neutrophils and mononuclear cells, and for cytokines IP-10, MCP-1, MIP1 and TNF-α. Moreover, the inhibition dose dependency result was evident for KC, MIP-1 and TNF-α. The very promising outcomes emerged from a comparison of SET-M33 activity and Budesonide, namely the corticosteroid drug used in clinical practice to reduce lung inflammation and in the present study as a positive control. In particular, as regards IP-10, MIP-1α and TNF-α levels, SET-M33 proved even more active in a dose-dependent manner than Budesonide in inhibiting the inflammatory effect provoked by LPS. 

At the same time, the in vivo toxicity study showed that doses compatible with anti-inflammatory efficacy could be considered suitable for clinical use. As stated by Palazzi et al. [47], changes likely to impair the functional capacity of an organism or degenerative changes associated with inflammation and fibrosis can be considered adverse. For these evaluations, we initially estimated the concentration of peptide in the aerosol, the particle-size distribution in the aerosol, and the estimated achieved dose of the peptide inhaled by the mice. This allowed us to calculate that doses of 5 and 20 mg/kg/day corresponded to actual concentrations in aerosol particles of 4.34 and 23.1 mg/kg/day, respectively. These data are important for evaluation of the actual number of molecules inhaled by the animals. The high incidence, nature and range of findings seen in mice treated with SET-M33 at 23.1 mg/kg/day indicated irritation and foreign-body reaction and included a clear local inflammatory response, leading to the conclusion that effects were adverse at this exposure level. On the other hand, pathological effects on the respiratory system were practically absent in mice dosed with 4.34 mg/kg/day, which did not provoke any in-life effects. As such, the effects were considered non-adverse. Consequently, the present results suggest a reference NOAEL for aerosol administration of 5 mg/kg as a starting point for future clinical studies.

The data reported in this article, added to that previously published for different animal models [28,29,36,37] complete the experimentation of the SET-M33 peptide. SET-M33 has now concluded all preclinical stages of development, including safety evaluations and toxicokinetic parameters, for administration by intravenous bolus or infusion [36] and inhalation.

## Figures and Tables

**Figure 1 ijms-24-07967-f001:**
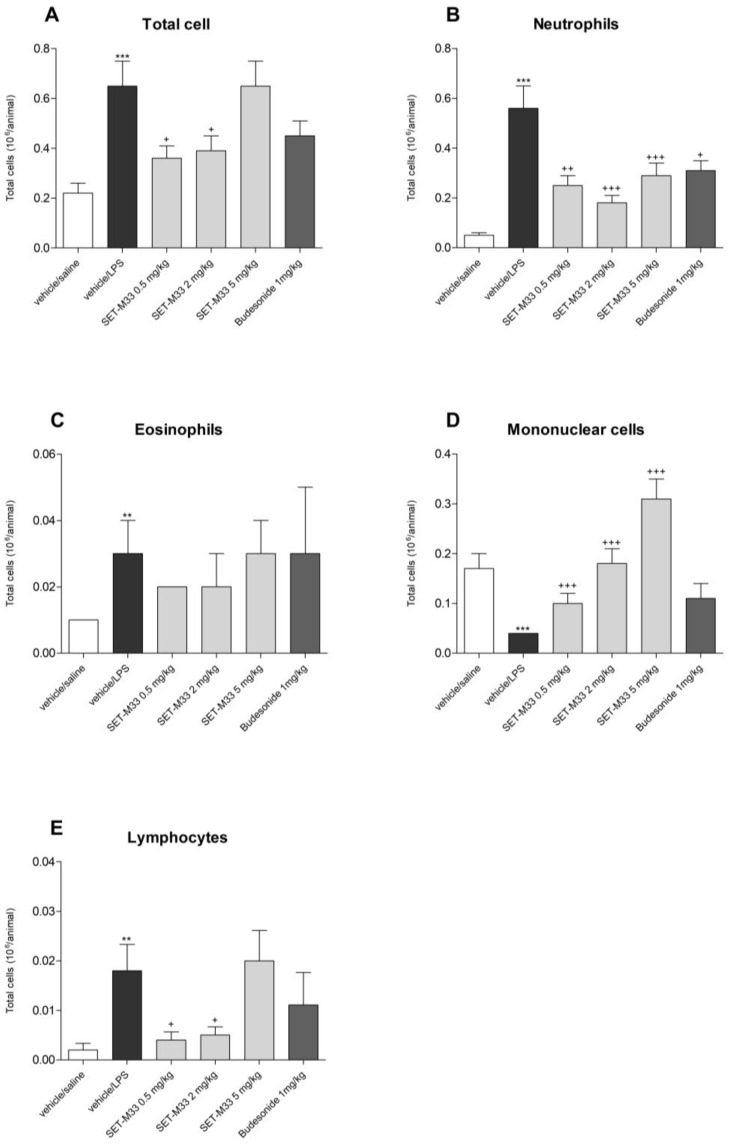
Effect of vehicle, SET-M33 (0.5, 2 and 5 mg/kg) and budesonide (1 mg/kg) on BAL total cell (**A**), neutrophils (**B**), eosinophils (**C**), mononuclear cells (**D**) and lymphocytes (**E**) (group mean values ± s.e.m.) in LPS-induced lung inflammation. Significant differences between groups are indicated as follows: *** *p* < 0.001 and ** *p* < 0.01 Vehicle/LPS versus Vehicle/Saline; +++ *p* < 0.001, ++ *p* < 0.01 and + *p* < 0.05 drug-treated groups versus Vehicle/LPS. Groups were compared by the Student’s *t* test. The graphs were plotted using GraphPad Prism for Windows version 5.03, GraphPad Software, San Diego, CA, USA.

**Figure 2 ijms-24-07967-f002:**
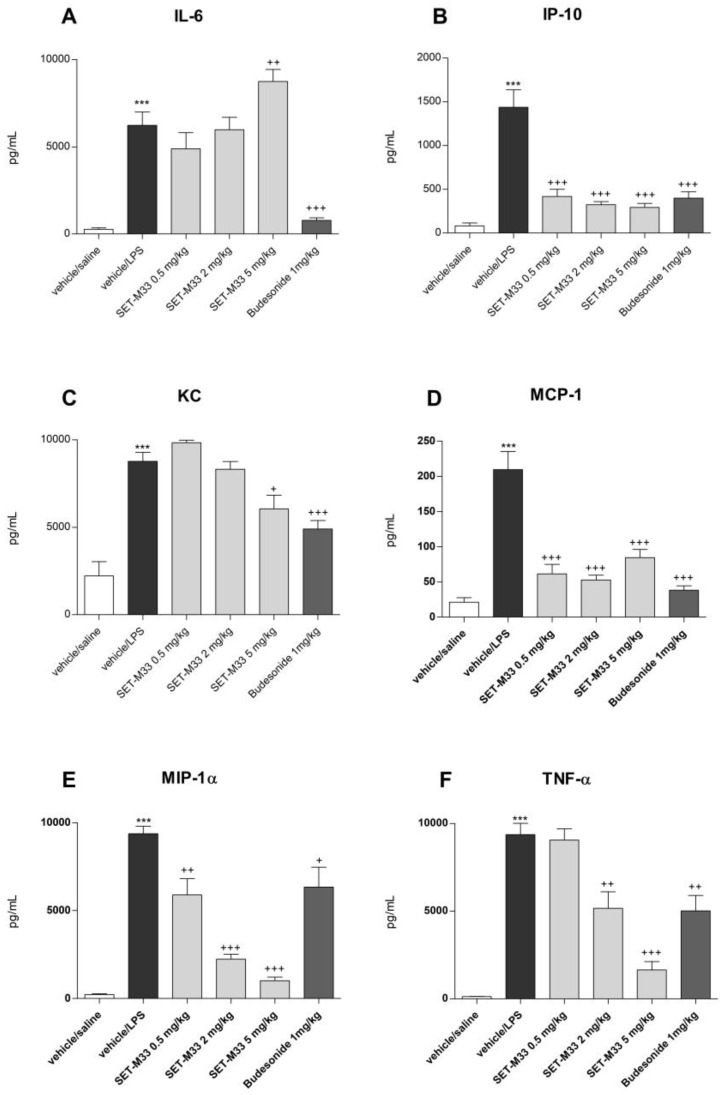
Effect of vehicle, SET-M33 (0.5, 2 and 5 mg/kg) and budesonide (1 mg/kg) on BAL levels of cytokines IL-6 (**A**), IP-10 (**B**), KC (**C**), MPC-1 (**D**), MIP-1α (**E**) and TNF-α (**F**) (group mean values ± s.e.m.) in LPS-induced lung inflammation. Significant differences between groups are indicated as follows: *** *p* < 0.001 Vehicle/ LPS versus Vehicle/Saline; +++ *p* < 0.001, ++ *p* < 0.01 and + *p* < 0.05 drug-treated groups versus Vehicle/LPS. Groups were compared by the Student’s *t* test. The graphs were plotted using GraphPad Prism for Windows version 5.03, GraphPad Software, San Diego, CA, USA.

**Table 1 ijms-24-07967-t001:** Atmosphere analysis and estimated inhaled dose of SET-M33 peptide achieved at 5 and 20 mg/kg/day inhaled by CD-1 mice.

Concentration (µg/L)			Particle Size		Dose (mg/kg/Day)	
Target	Achieved		MMAD (µm)	σg	Target	Estimated Achieved Inhaled
	Mean	SD				
79	68.5	17.8	0.8	2.3	5	4.34
318	371	25.4	1.4	2.5	20	23.1

Abbreviation: SD = Standard deviation; MMAD = Mass Median Aerodynamic Diameters; σg = geometric standard deviation.

**Table 2 ijms-24-07967-t002:** SET-M33-related findings in the lungs, nose/turbinates, larynx and tracheal bifurcation in CD-1 mice after treatment with inhaled peptide at 0, 4.34 and 23.1 mg/kg/day for 1 week. The findings were assigned a severity grade: minimal, slight, moderate, marked or severe.

		Male			Female		
		0 mg/kg/Day	4.34 mg/kg/Day	23.1 mg/kg/Day	0 mg/kg/Day	4.34 mg/kg/Day	23.1 mg/kg/Day
*SET-M33-related findings in the lungs*							
Inflammation, interstitial	Minimal	0	0	1	1	2	4
	Slight	0	0	4	0	0	2
	Total	0	0	5	1	2	6
Inflammation, granulomatous	Minimal	0	0	2	0	0	3
	Slight	0	0	3	0	0	0
	Total	0	0	5	0	0	3
Infiltrate, inflammatory cell, perivascular	Minimal	0	0	2	0	1	1
	Slight	0	0	0	0	0	1
	Total	0	0	2	0	1	2
Inflammation, alveolar ducts	Minimal	0	0	2	1	0	2
	Total	0	0	2	1	0	2
Fibrosis, alveolar ducts	Minimal	0	0	0	0	0	1
	Total	0	0	0	0	0	1
*SET-M33-related findings in the nose/turbinates*							
Atrophy/degeneration, olfactory epithelium	Minimal	0	2	1	1	4	0
	Slight	0	4	5	0	2	3
	Moderate	0	0	0	0	0	3
	Total	0	6	6	1	6	6
Inflammatory exudate	Minimal	0	4	0	0	0	2
	Slight	0	0	2	0	0	0
	Moderate	0	0	3	0	0	3
	Marked	0	0	0	0	0	1
	Total	0	4	5	0	0	6
Eosinophilic globules, olfactory epithelium	Minimal	0	6	3	0	6	3
	Total	0	6	3	0	6	3
Eosinophilic globules, respiratory epithelium	Minimal	0	3	3	0	6	2
	Total	0	3	3	0	6	2
Infiltrate, inflammatory cell, lamina propria	Minimal	0	1	2	0	3	2
	Slight	0	1	0	0	2	2
	Moderate	0	0	0	0	0	1
	Total	0	2	2	0	5	5
Haemorrhage	Moderate	0	0	1	0	0	0
	Total	0	0	1	0	0	0
*SET-M33-related findings in the larynx*							
Squamous metaplasia	Minimal	0	4	6	0	6	6
	Total	0	4	6	0	6	6
Infiltrate, inflammatory cell	Minimal	0	3	4	0	0	5
	Total	0	3	4	0	0	5
*SET-M33-related findings at the tracheal bifurcation*							
Loss of cilia, point of bifurcation	Minimal	0	0	2	1	0	3
	Total	0	0	2	1	0	3

## Data Availability

Not applicable.

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
