# Peer review of "In Vivo Efficacy and Toxicity of an Antimicrobial Peptide in a Model of Endotoxin-Induced Pulmonary Inflammation"

_ijms, 2023, doi:10.3390/ijms24097967_

Round 1

Reviewer 1 Report

This manuscript by Cresti et al. describes “In-vivo efficacy and toxicity of an antimicrobial peptide in a model of endotoxin-induced pulmonary inflammation”.  This work is well planned, results and discussions sections are well written in this manuscript. Authors nicely explored in-vivo efficacy and toxicity of synthetic peptide SET-M33 in murine model of pulmonary inflammation. I would recommend this manuscript may be considered for publication after corrections as below.

1.     As this manuscript is related to activity of peptide, I would recommend authors to include a paragraph about peptide drug discovery in introduction.

2.     Figure 1 and figure 2 are looking same in this manuscript. Please upload figure 2 containing effect of this peptide on BAL cytokine levels.

3.      Authors can write in detail about specific inhalation exposure system for mice in section 3.2

Reviewer 2 Report

Author needs to improve much in Introduction, result and discussion section. The result image for cytokine is missing/ mismatch, required some deep discussion about the result.

Line 67-68; Dose selection for inhalation is not appreciable through different route administered study. For inhalation, required range finding study data as a preliminary study. This needs to be scientific justification with reference.

Line 71; Mention as Preliminary atmosphere analysis for the SET M33 peptide exposure without animals needs to present details and data table.

Table 1; The ideal range was not attained in 79 μg/L from this the relevant dose of SET M33 was not reached to the regions of lung. kindly justify with the scientific reasoning.

Table 1, Line 90; The standard deviation value is high

Line 107; mention the clinical signs specifically for unsteadiness with gait or tremor, etc.

Line 113; mention whether it is weekly feed consumption or else.

Line 118; Mention the statistical significance for the organ weight. Also derive the Relative organ weight with brain and show the significance.

Line 207; Figure 2, the image presented is not about the Cytokine data.

Line 215; Regarding materials such as SET-M33, LPS, Budesonide, isoflurane, Tween 80, TFA, Pentobarbital sodium, LI-heparin, Inhalation unit detail, UPLC instrument detail.

Line 232; Mention the chamber condition during exposure. How the chamber condition was assessed at what interval. Also, the animal acclimatization procedure the chamber needs to brief.

Line 234; Replace word of ”administration” with ”exposure”.

Line 237; 50uL of which dose administered, specify LPS or other... And mention the concentration used for the administration

Line 285; Mention IP pentobarbital dose used for killing animals.

Line 302; Instrument detail for LCMS/MS required.

Line 327; Multiplex system detail required.

Line 357; the SET-M33 need to mention correctly.

Line 414; More recent reference needs to cite for the references.

Round 2

Reviewer 2 Report

Author required to care on units, maintain uniformity for kit/product purchase with catalogue number and make. Also reference needs to check in one of the reference mention [124-26]. Few comments still have not response.

Author Response

We thank the referee for the suggestion. Now, in material and methods we added all the information about kit/product code numbers and make. Moreover, the reference mistake number (24 and not 124) is now corrected.